# The Price of Fair PCA: One Extra Dimension

**Samira Samadi**
Georgia Tech
ssamadi6@gatech.edu

**Uthaipon Tantipongpipat**
Georgia Tech
tao@gatech.edu

**Jamie Morgenstern**
Georgia Tech
jamiemmt.cs@gatech.edu

**Mohit Singh**
Georgia Tech
mohitsinghr@gmail.com

**Santosh Vempala**
Georgia Tech
vempala@cc.gatech.edu

## Abstract

We investigate whether the standard dimensionality reduction technique of PCA inadvertently produces data representations with different fidelity for two different populations. We show on several real-world data sets, PCA has higher reconstruction error on population $A$ than on $B$ (for example, women versus men or lower- versus higher-educated individuals). This can happen even when the data set has a similar number of samples from $A$ and $B$. This motivates our study of dimensionality reduction techniques which maintain similar fidelity for $A$ and $B$. We define the notion of Fair PCA and give a polynomial-time algorithm for finding a low dimensional representation of the data which is nearly-optimal with respect to this measure. Finally, we show on real-world data sets that our algorithm can be used to efficiently generate a fair low dimensional representation of the data.

## 1 Introduction

In recent years, the ML community has witnessed an onslaught of charges that real-world machine learning algorithms have produced "biased" outcomes. The examples come from diverse and impactful domains. Google Photos labeled African Americans as gorillas [Twitter, 2015; Simonite, 2018] and returned queries for CEOs with images overwhelmingly male and white [Kay et al., 2015], searches for African American names caused the display of arrest record advertisements with higher frequency than searches for white names [Sweeney, 2013], facial recognition has wildly different accuracy for white men than dark-skinned women [Buolamwini and Gebru, 2018], and recidivism prediction software has labeled low-risk African Americans as high-risk at higher rates than low-risk white people [Angwin et al., 2018].

The community's work to explain these observations has roughly fallen into either "biased data" or "biased algorithm" bins. In some cases, the training data might under-represent (or over-represent) some group, or have noisier labels for one population than another, or use an imperfect proxy for the prediction label (e.g., using arrest records in lieu of whether a crime was committed). Separately, issues of imbalance and bias might occur due to an algorithm's behavior, such as focusing on accuracy across the entire distribution rather than guaranteeing similar false positive rates across populations, or by improperly accounting for confirmation bias and feedback loops in data collection. If an algorithm fails to distribute loans or bail to a deserving population, the algorithm won't receive additional data showing those people would have paid back the loan, but it will continue to receive more data about the populations it (correctly) believed should receive loans or bail.

Many of the proposed solutions to "biased data" problems amount to re-weighting the training set or adding noise to some of the labels; for "biased algorithms", most work has focused on maximizing accuracy subject to a constraint forbidding (or penalizing) an unfair model. Both of these concerns

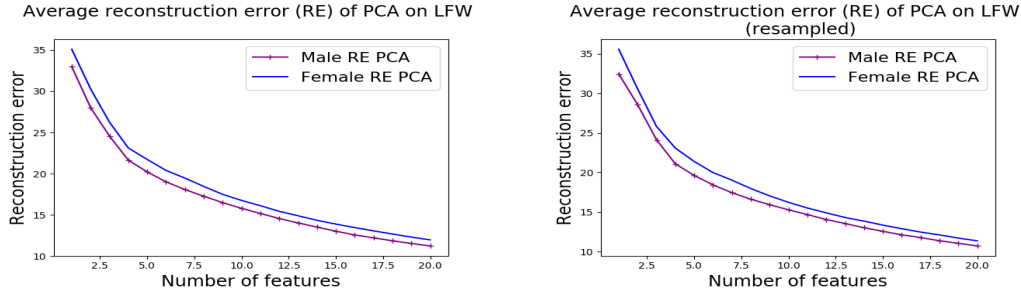

Figure 1: Left: Average reconstruction error of PCA on labeled faces in the wild data set (LFW), separated by gender. Right: The same, but sampling 1000 faces with men and women equiprobably (mean over 20 samples).

and approaches have significant merit, but form an incomplete picture of the ML pipeline and where unfairness might be introduced therein. Our work takes another step in fleshing out this picture by analyzing when *dimensionality reduction* might inadvertently introduce bias. We focus on principal component analysis (henceforth PCA), perhaps the most fundamental dimensionality reduction technique in the sciences [Pearson, 1901; Hotelling, 1933; Jolliffe, 1986]. We show several real-world data sets for which PCA incurs much higher average reconstruction error for one population than another, even when the populations are of similar sizes. Figure 1 shows that PCA on labeled faces in the wild data set (LFW) has higher reconstruction error for women than men even if male and female faces are sampled with equal weight.

This work underlines the importance of considering fairness and bias at every stage of data science, not only in gathering and documenting a data set [Gebru et al., 2018] and in training a model, but also in any interim data processing steps. Many scientific disciplines have adopted PCA as a default preprocessing step, both to avoid the curse of dimensionality and also to do exploratory/explanatory data analysis (projecting the data into a number of dimensions that humans can more easily visualize). The study of human biology, disease, and the development of health interventions all face both aforementioned difficulties, as do numerous economic and financial analysis. In such high-stakes settings, where statistical tools will help in making decisions that affect a diverse set of people, we must take particular care to ensure that we share the benefits of data science with a diverse community.

We also emphasize this work has implications for representational rather than just allocative harms, a distinction drawn by Crawford [2017] between how people are represented and what goods or opportunities they receive. Showing primates in search results for African Americans is repugnant primarily due to its representing and reaffirming a racist painting of African Americans, not because it directly reduces any one person's access to a resource. If the default template for a data set begins with running PCA, and PCA does a better job representing men than women, or white people over minorities, the new representation of the data set itself may rightly be considered an unacceptable sketch of the world it aims to describe.

Our work proposes a different linear dimensionality reduction which aims to represent two populations $A$ and $B$ with similar *fidelity*—which we formalize in terms of *reconstruction error*. Given an $n$-dimensional data set and its $d$-dimensional approximation, the reconstruction error of the data with respect to its low-dimensional approximation is the sum of squares of distances between the original data points and their approximated points in the $d$-dimensional subspace. To eliminate the effect of size of a population, we focus on average reconstruction error over a population. One possible objective for our goal would find a $d$-dimensional approximation of the data which minimizes the maximum reconstruction error over the two populations. However, this objective doesn't avoid grappling with the fact that population $A$ may perfectly embed into $d$ dimensions, whereas $B$ might require many more dimensions to have low reconstruction error. In such cases, this objective would not necessarily favor a solution with average reconstruction error of $\epsilon$ for $A$ and $y \gg \epsilon$ for $B$ over one with $y$ error for $A$ and $y$ error for $B$. This holds even if $B$ requires $y$ reconstruction error to be embedded into $d$ dimensions and thus the first solution is nearly optimal for both populations in $d$ dimensions.

This motivates our focus on finding a projection which minimizes the maximum *additional* or *marginal* reconstruction error for each population above the optimal $n$ into $d$ projection for that population alone. This quantity captures how much a population's reconstruction error increases by including another population in the dimensionality reduction optimization. Despite this computational problem appearing more difficult than solving "vanilla" PCA, we introduce a polynomial-time algorithm which finds an $n$ into $(d + 1)$-dimensional embedding with objective value better than any $d$-dimensional embedding. Furthermore, we show that optimal solutions have equal additional average error for populations $A$ and $B$.

**Summary of our results**  We show PCA can overemphasize the reconstruction error for one population over another (equally sized) population, and we should therefore think carefully about dimensionality reduction in domains where we care about fair treatment of different populations. We propose a new dimensionality reduction problem which focuses on representing $A$ and $B$ with similar additional error over projecting $A$ or $B$ individually. We give a polynomial-time algorithm which finds near-optimal solutions to this problem. Our algorithm relies on solving a semidefinite program (SDP), which can be prohibitively slow for practical applications. We note that it is possible to (approximately) solve an SDP with a much faster multiplicative-weights style algorithm, whose running time in practice is equivalent to solving standard PCA at most 10-15 times. The details of the algorithm are given in the full version of this work. We then evaluate the empirical performance of this algorithm on several human-centric data sets.

## 2   Related work

This work contributes to the area of fairness for machine learning models, algorithms, and data representations. One interpretation of our work is that we suggest using Fair PCA, rather than PCA, when creating a lower-dimensional representation of a data set for further analysis. Both pieces of work which are most relevant to our work take the posture of explicitly trying to reduce the correlation between a sensitive attribute (such as race or gender) and the new representation of the data. The first piece is a broad line of work [Zemel et al., 2013; Beutel et al., 2017; Calmon et al., 2017; Madras et al., 2018; Zhang et al., 2018] that aims to design representations which will be conditionally independent of the protected attribute, while retaining as much information as possible (and particularly task-relevant information for some fixed classification task). The second piece is the work by Olfat and Aswani [2018], who also look to design PCA-like maps which reduce the projected data's dependence on a sensitive attribute. Our work has a qualitatively different goal: we aim not to hide a sensitive attribute, but instead to maintain as much information about each population after projecting the data. In other words, we look for representation with similar richness for population $A$ as $B$, rather than making $A$ and $B$ indistinguishable.

Other work has developed techniques to obfuscate a sensitive attribute directly [Pedreshi et al., 2008; Kamiran et al., 2010; Calders and Verwer, 2010; Kamiran and Calders, 2011; Luong et al., 2011; Kamiran et al., 2012; Kamishima et al., 2012; Hajian and Domingo-Ferrer, 2013; Feldman et al., 2015; Zafar et al., 2015; Fish et al., 2016; Adler et al., 2016]. This line of work diverges from ours in two ways. First, these works focus on representations which obfuscate the sensitive attribute rather than a representation with high fidelity regardless of the sensitive attribute. Second, most of these works do not give formal guarantees on how much an objective will degrade after their transformations. Our work directly minimizes the amount by which each group's marginal reconstruction error increases.

Much of the other work on fairness for learning algorithms focuses on fairness in classification or scoring [Dwork et al., 2012; Hardt et al., 2016; Kleinberg et al., 2016; Chouldechova, 2017], or online learning settings [Joseph et al., 2016; Kannan et al., 2017; Ensign et al., 2017b,a]. These works focus on either statistical parity of the decision rule, or equality of false positives or negatives, or an algorithm with a fair decision rule. All of these notions are driven by a single learning task rather than a generic transformation of a data set, while our work focuses on a ubiquitous, task-agnostic preprocessing step.

## 3   Notation and vanilla PCA

We are given $n$-dimensional data points represented as rows of matrix $M \in \mathbb{R}^{m \times n}$. We will refer to the *set* and *matrix* representation interchangeably. The data consists of two subpopulations $A$ and

$B$ corresponding to two groups with different value of a binary sensitive attribute (e.g., males and females). We denote by $\begin{bmatrix} A \\ B \end{bmatrix}$ the concatenation of two matrices $A, B$ by row. We refer to the $i^{th}$ row of $M$ as $M_i$, the $j^{th}$ column of $M$ as $M^j$ and the $(i, j)^{th}$ element of $M$ as $M_{ij}$. We denote the Frobenius norm of matrix $M$ by $\|M\|_F$ and the 2-norm of the vector $M_i$ by $\|M_i\|$. For $k \in \mathbb{N}$, we write $[k] := \{1, \dots, k\}$. $|A|$ denotes the size of a set $A$. Given two matrices $M$ and $N$ of the same size, the Frobenius inner product of these matrices is defined as $\langle M, N \rangle = \sum_{ij} M_{ij} N_{ij} = \text{Tr}(M^T N)$.

## 3.1 PCA

This section recalls useful facts about PCA that we use in later sections. We begin with a reminder of the definition of the PCA problem in terms of minimizing the reconstruction error of a data set.

**Definition 3.1.** *(PCA problem) Given a matrix $M \in \mathbb{R}^{m \times n}$, find a matrix $\widehat{M} \in \mathbb{R}^{m \times n}$ of rank at most $d$ $(d \leq n)$ that minimizes $\|M - \widehat{M}\|_F$.*

We will refer to $\widehat{M}$ as an optimal rank-$d$ approximation of $M$. The following well-known fact characterizes the solutions to this classic problem [e.g., Shalev-Shwartz and Ben-David, 2014].

**Fact 3.1.** *If $\widehat{M}$ is a solution to the PCA problem, then $\widehat{M} = MWW^T$ for a matrix $W \in \mathbb{R}^{n \times d}$ with $W^T W = I$. The columns of $W$ are eigenvectors corresponding to top $d$ eigenvalues of $M^T M$.*

The matrix $WW^T \in \mathbb{R}^{n \times n}$ is called a projection matrix.

## 4 Fair PCA

Given the $n$-dimensional data with two subgroups $A$ and $B$, let $\widehat{M}, \widehat{A}, \widehat{B}$ be optimal rank-$d$ PCA approximations for $M, A$, and $B$, respectively. We introduce our approach to fair dimensionality reduction by giving two compelling examples of settings where dimensionality reduction inherently makes a tradeoff between groups $A$ and $B$. Figure 2a shows a setting where projecting onto any single dimension either favors $A$ or $B$ (or incurs significant reconstruction error for both), while either group separately would have a high-fidelity embedding into a single dimension. This example suggests any projection will necessarily make a trade off between error on $A$ and error on $B$.

Our second example (shown in Figure 2b) exhibits a setting where $A$ and $B$ suffer very different reconstruction error when projected onto one dimension: $A$ has high reconstruction error for every projection while $B$ has a perfect representation in the horizontal direction. Thus, asking for a projection which minimizes the maximum reconstruction error for groups $A$ and $B$ might require incurring additional error for $B$ while not improving the error for $A$. So, minimizing the maximum reconstruction error over $A$ and $B$ fails to account for the fact that two populations might have wildly different representation error when embedded into $d$ dimensions. Optimal solutions to such objective might behave in a counterintuitive way, preferring to exactly optimize for the group with larger inherent representation error rather than approximately optimizing for both groups simultaneously. We find this behaviour undesirable—it requires sacrifice in quality for one group for no improvement for the other group.

**Remark 4.1.** *We focus on the setting where we ask for a single projection into $d$ dimensions rather than two separate projections because using two distinct projections (or more generally two models) for different populations raises legal and ethical concerns. Learning two different projections also faces no inherent tradeoff in representing $A$ or $B$ with those projections.*[1]

We therefore turn to finding a projection which minimizes the maximum deviation of each group from its optimal projection. This optimization asks that $A$ and $B$ suffer a similar *loss* for being projected together into $d$ dimensions compared to their individually optimal projections. We now introduce our notation for measuring a group's loss when being projected to $Z$ rather than to its optimal $d$-dimensional representation:

**Definition 4.2** (Reconstruction error). *Given two matrices $Y$ and $Z$ of the same size, the reconstruction error of $Y$ with respect to $Z$ is defined as*

$$error(Y, Z) = \|Y - Z\|_F^2.$$

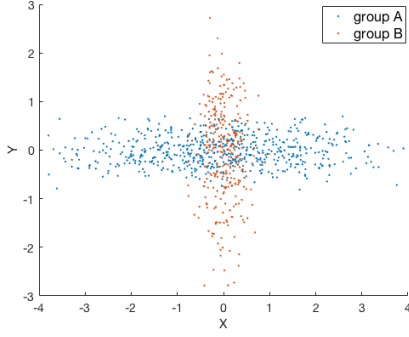
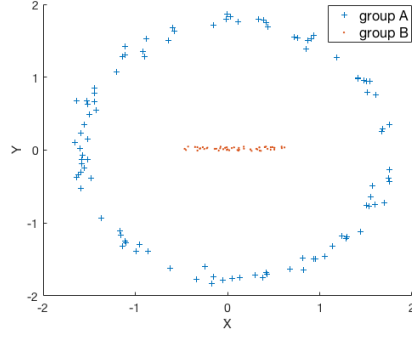

(a) The best one dimensional PCA projection for group $A$ is vector $(1,0)$ and for group $B$ it is vector $(0,1)$.

(b) Group $B$ has a perfect one-dimensional projection. For group $A$, any one-dimensional projection is equally bad.

Figure 2

**Definition 4.3** (Reconstruction loss)**.** *Given a matrix $Y \in \mathbb{R}^{a \times n}$, let $\widehat{Y} \in \mathbb{R}^{a \times n}$ be the optimal rank-$d$ approximation of $Y$. For a matrix $Z \in \mathbb{R}^{a \times n}$ with rank at most $d$ we define*

$$loss(Y, Z) := \|Y - Z\|_F^2 - \|Y - \widehat{Y}\|_F^2.$$

Then, the optimization that we study asks to minimize the maximum loss suffered by any group. This captures the idea that, fixing a feasible solution, the objective will only improve if it improves the loss for the group whose current representation is worse. Furthermore, considering the reconstruction loss and not the reconstruction error prevents the optimization from incurring error for one subpopulation without improving the error for the other one as described in Figure 2b.

**Definition 4.4** (Fair PCA)**.** *Given $m$ data points in $\mathbb{R}^n$ with subgroups $A$ and $B$, we define the problem of finding a fair PCA projection into $d$-dimensions as optimizing*

$$\min_{U \in \mathbb{R}^{m \times n}, \; \mathsf{rank}(U) \leq d} \max \left\{ \frac{1}{|A|} loss(A, U_A), \frac{1}{|B|} loss(B, U_B) \right\}, \tag{1}$$

*where $U_A$ and $U_B$ are matrices with rows corresponding to rows of $U$ for groups $A$ and $B$ respectively.*

This definition does not appear to have a closed-form solution (unlike vanilla PCA—see Fact 3.1). To take a step in characterizing solutions to this optimization, Theorem 4.5 states that a fair PCA low dimensional approximation of the data results in the same loss for both groups.

**Theorem 4.5.** *Let $U$ be a solution to the Fair PCA problem* (1)*, then*

$$\frac{1}{|A|} loss(A, U_A) = \frac{1}{|B|} loss(B, U_B).$$

Before proving Theorem 4.5, we need to state some building blocks of the proof, Lemmas 4.6, 4.7, and 4.8. For the proofs of the lemmas please refer to the appendix B.

**Lemma 4.6.** *Given a matrix $U \in \mathbb{R}^{m \times n}$ such that $\mathsf{rank}(U) \leq d$, let $f(U) = \max \left\{ \frac{1}{|A|} loss(A, U_A), \frac{1}{|B|} loss(B, U_B) \right\}$. Let $\{v_1, \ldots, v_d\} \subset \mathbb{R}^n$ be an orthonormal basis of the row space of $U$ and $V := [v_1, \ldots, v_d] \in \mathbb{R}^{n \times d}$. Then*

$$f\left( \begin{bmatrix} A \\ B \end{bmatrix} VV^T \right) = f\left( \begin{bmatrix} AVV^T \\ BVV^T \end{bmatrix} \right) \leq f(U).$$

The next lemma presents some equalities that we will use frequently in the proofs.

**Lemma 4.7.** *Given a matrix $V = [v_1, \ldots, v_d] \in \mathbb{R}^{n \times d}$ with orthonormal columns, we have:*

$$\diamond \; loss(A, AVV^T) = \|\widehat{A}\|_F^2 - \textstyle\sum_{i=1}^d \|Av_i\|^2 = \|\widehat{A}\|_F^2 - \langle A^T A, VV^T \rangle$$

$\diamond \ \|A - AVV^T\|_F^2 = \|A\|_F^2 - \|AV\|_F^2 = \|A\|_F^2 - \sum_{i=1}^{d} \|Av_i\|^2$

Let the function $g_A = g_A(U)$ measure the reconstruction error of a fixed matrix $A$ with respect to its orthogonal projection to the input subspace $U$. The next lemma shows that the value of the function $g_A$ at any local minimum is the same.

**Lemma 4.8.** *Given a matrix $A \in \mathbb{R}^{a \times n}$, and a $d$-dimensional subspace $U$, let the function $g_A = g_A(U)$ denote the reconstruction error of matrix $A$ with respect to its orthogonal projection to the subspace $U$, that is $g_A(U) := \|A - AUU^T\|_F^2$, where by abuse of notation we use $U$ inside the norm to denote the matrix which has an orthonormal basis of the subspace $U$ as its columns. The value of the function $g_A$ at any local minimum is the same.*

**Proof of Theorem 4.5**:

Consider the functions $g_A$ and $g_B$ defined in Lemma 4.8. It follows from Lemma 4.6 and Lemma 4.7 that for $V \in \mathbb{R}^{n \times d}$ with $V^T V = I$ we have

$$loss(A, AVV^T) = \|\widehat{A}\|_F^2 - \|A\|_F^2 + g_A(V), \tag{2}$$
$$loss(B, BVV^T) = \|\widehat{B}\|_F^2 - \|B\|_F^2 + g_B(V).$$

Therefore, the Fair PCA problem is equivalent to

$$\min_{V \in \mathbb{R}^{n \times d}, V^T V = I} f(V) := \max \left\{ \frac{1}{|A|} loss(A, AVV^T), \frac{1}{|B|} loss(B, BVV^T) \right\}.$$

We proceed to prove the claim by contradiction. Let $W$ be a global minimum of $f$ and assume that

$$\frac{1}{|A|} loss(A, AWW^T) > \frac{1}{|B|} loss(B, BWW^T). \tag{3}$$

Hence, since $loss$ is continuous, for any matrix $W_\epsilon$ with $W_\epsilon^T W_\epsilon = I$ in a small enough neighborhood of $W$, $f(W_\epsilon) = \frac{1}{|A|} loss(A, AW_\epsilon W_\epsilon^T)$. Since $W$ is a global minimum of $f$, it is a local minimum of $\frac{1}{|A|} loss(A, AWW^T)$ or equivalently a local minimum of $g_A$ because of (2).

Let $\{v_1, \ldots, v_n\}$ be an orthonormal basis of the eigenvectors of $A^T A$ corresponding to eigenvalues $\lambda_1 \geq \lambda_2 \geq \ldots \geq \lambda_n$. Let $V^*$ be the subspace spanned by $\{v_1, \ldots, v_d\}$. Note that $loss(A, AV^{*T}V^*) = 0$. Since the loss is always non-negative for both $A$ and $B$, (3) implies that $loss(A, AWW^T) > 0$. Therefore, $W \neq V^*$ and $g_A(V^*) < g_A(W)$. By Lemma 4.8, this is in contradiction with $V^*$ being a global minimum and $W$ being a local minimum of $g_A$. $\qquad \square$

# 5 Algorithm and analysis

In this section, we present a polynomial-time algorithm for solving the fair PCA problem. Our algorithm outputs a matrix of rank at most $d + 1$ and guarantees that it achieves the fair PCA objective value equal to the optimal $d$-dimensional fair PCA value. The algorithm has two steps: first, relax fair PCA to a semidefinite optimization problem and solve the SDP; second, solve an LP designed to reduce the rank of said solution. We argue using properties of extreme point solutions that the solution must satisfy a number of constraints of the LP with equality, and argue directly that this implies the solution must lie in $d + 1$ or fewer dimensions. We refer the reader to Lau et al. [2011] for basics and applications of this technique in approximation algorithms.

**Theorem 5.1.** *There is a polynomial-time algorithm that outputs an approximation matrix of the data such that it is either of rank $d$ and is an optimal solution to the fair PCA problem OR it is of rank $d + 1$, has equal losses for the two populations and achieves the optimal fair PCA objective value for dimension $d$.*

**Proof of Theorem 5.1**: The algorithm to prove Theorem 5.1 is presented in Algorithm 1. Using Lemma 4.7, we can write the semi-definite relaxation of the fair PCA objective (Def. 4.4) as SDP (4). This semi-definite program can be solved in polynomial time. The system of constraints (5)-(9) is a

**Algorithm 1:** Fair PCA

---

**Input** : $A \in \mathbb{R}^{m_1 \times n}, B \in \mathbb{R}^{m_2 \times n}, d < n, m = m_1 + m_2$
**Output**: $U \in \mathbb{R}^{m \times n}, \mathsf{rank}(U) \leq d + 1$

1 Find optimal rank-$d$ approximations of $A, B$ as $\widehat{A}, \widehat{B}$ (e.g. by Singular Value Decomposition).
2 Let $(\hat{P}, \hat{z})$ be a solution to the SDP:

$$\min_{P \in \mathbb{R}^{n \times n}, z \in \mathbb{R}} \quad z \tag{4}$$

$$\text{s.t. } z \geq \frac{1}{m_1} \cdot \left( \|\widehat{A}\|_F^2 - \langle A^\top A, P \rangle \right)$$

$$z \geq \frac{1}{m_2} \cdot \left( \|\widehat{B}\|_F^2 - \langle B^\top B, P \rangle \right)$$

$$\mathsf{Tr}(P) \leq d, \; 0 \preceq P \preceq I$$

3 Apply Singular Value Decomposition to $\hat{P}$, $\hat{P} = \sum_{j=1}^n \hat{\lambda}_j u_j u_j^T$.
4 Find an extreme solution $(\bar{\lambda}, z^*)$ of the LP:

$$\min_{\lambda \in \mathbb{R}^n, z \in \mathbb{R}} \quad z \tag{5}$$

$$\text{s.t. } z \geq \frac{1}{m_1} \left( \|\widehat{A}\|_F^2 - \langle A^\top A, \sum_{j=1}^n \lambda_j u_j u_j^T \rangle \right) = \frac{1}{m_1} \left( \|\widehat{A}\|_F^2 - \sum_{j=1}^n \lambda_j \cdot \langle A^\top A, u_j u_j^T \rangle \right) \tag{6}$$

$$z \geq \frac{1}{m_2} \left( \|\widehat{B}\|_F^2 - \langle B^\top B, \sum_{j=1}^n \lambda_j u_j u_j^T \rangle \right) = \frac{1}{m_2} \left( \|\widehat{B}\|_F^2 - \sum_{j=1}^n \lambda_j \cdot \langle B^\top B, u_j u_j^T \rangle \right) \tag{7}$$

$$\sum_{i=1}^n \lambda_i \leq d \tag{8}$$

$$0 \leq \lambda_i \leq 1 \tag{9}$$

5 Set $P^* = \sum_{j=1}^n \lambda_j^* u_j u_j^T$ where $\lambda_j^* = 1 - \sqrt{1 - \bar{\lambda}_j}$.
6 **return** $U = \begin{bmatrix} A \\ B \end{bmatrix} P^*$

---

linear program in the variables $\lambda_i$ (with the $u_i$'s fixed). Therefore, an extreme point solution $(\bar{\lambda}, z^*)$ is defined by $n + 1$ equalities, at most three of which can be constraints in (6)-(8) and the rest (at least $n - 2$ of them) must be from the $\bar{\lambda}_i = 0$ or $\bar{\lambda}_i = 1$ for $i \in [n]$. Given the upper bound of $d$ on the sum of the $\bar{\lambda}_i$'s, this implies that at least $d - 1$ of them are equal to 1, i.e., at most two are fractional and add up to 1.

**Case 1.** All the eigenvalues are integral. Therefore, there are $d$ eigenvalues equal to 1. This results in orthogonal projection to $d$-dimension.

**Case 2.** $n - 2$ of eigenvalues are in $\{0, 1\}$ and two eigenvalues $0 < \bar{\lambda}_d, \bar{\lambda}_{d+1} < 1$. Since we have $n + 1$ tight constraints, this means that both of the first two constraints are tight. Therefore

$$\frac{1}{|A|}(\|\widehat{A}\|_F^2 - \sum_{i=1}^n \bar{\lambda}_i \langle A^T A, u_i u_i^T \rangle) = \frac{1}{|B|}(\|\widehat{B}\|_F^2 - \sum_{i=1}^n \bar{\lambda}_i \langle B^T B, u_i u_i^T \rangle) = z^* \leq \hat{z},$$

where the inequality is by observing that $(\hat{\lambda}, \hat{z})$ is a feasible solution. Note that the loss of group $A$ given by an affine projection $P^* = \sum_{j=1}^n \lambda^* u_j u_j^T$ is

$$loss(A, AP^*) = \|A - AP^*\|_F^2 - \|A - \widehat{A}\|_F^2 = \mathsf{Tr}\left((A - AP^*)(A - AP^*)^\top\right) - \|A\|_F^2 + \|\widehat{A}\|_F^2$$

$$= \mathsf{Tr}\left((A - AP^*)(A - AP^*)^\top\right) - \|A\|_F^2 + \|\widehat{A}\|_F^2 = \|\widehat{A}\|_F^2 - 2\mathsf{Tr}(AP^* A^\top) + \mathsf{Tr}(AP^{*2} A^\top)$$

$$= \|\widehat{A}\|_F^2 - \sum_{i=1}^n (2\lambda_i^* - \lambda_i^{*2}) \langle A^T A, u_i u_i^T \rangle = \|\widehat{A}\|_F^2 - \sum_{i=1}^n \bar{\lambda} \langle A^T A, u_i u_i^T \rangle,$$

where the last inequality is by the choice of $\lambda_j^* = 1 - \sqrt{1 - \bar{\lambda}_j}$. The same equality holds true for group $B$. Therefore, $P^*$ gives the equal loss of $z^* \leq \hat{z}$ for two groups. The embedding $x \rightarrow (x \cdot u_1, \ldots, x \cdot u_{d-1}, \sqrt{\lambda_d^*} x \cdot u_d, \sqrt{\lambda_{d+1}^*} x \cdot u_{d+1})$ corresponds to the affine projection of any point (row) of $A, B$ defined by the solution $P^*$.

In both cases, the objective value is at most that of the original fairness objective. □

The result of Theorem 5.1 in two groups generalizes to more than two groups as follows. Given $m$ data points in $\mathbb{R}^n$ with $k$ subgroups $A_1, A_2, \ldots, A_k$, and $d \leq n$ the desired number of dimensions of projected space, we generalize Definition 4.4 of fair PCA problem as optimizing

$$\min_{U \in \mathbb{R}^{m \times n}, \, \text{rank}(U) \leq d} \max_{i \in \{1, \ldots, k\}} \left\{ \frac{1}{|A_i|} loss(A_i, U_{A_i}) \right\}, \tag{10}$$

where $U_{A_i}$ are matrices with rows corresponding to rows of $U$ for groups $A_i$.

**Theorem 5.2.** *There is a polynomial-time algorithm to find a projection such that it is of dimension at most $d + k - 1$ and achieves the optimal fairness objective value for dimension d.*

In contrast to the case of two groups, when there are more than two groups in the data, it is possible that all optimal solutions to fair PCA will not assign the same loss to all groups. However, with $k - 1$ extra dimensions, we can ensure that the loss of each group remains at most the optimal fairness objective in $d$ dimension. The result of Theorem 5.2 follows by extending algorithm in Theorem 5.1 by adding linear constraints to SDP and LP for each extra group. An extreme solution $(\bar{\lambda}, z^*)$ of the resulting LP contains at most $k$ of $\lambda_i$'s that are strictly in between 0 and 1. Therefore, the final projection matrix $P^*$ has rank at most $d + k - 1$.

**Runtime**  We now analyze the runtime of Algorithm 1, which consists of solving SDP (4) and finding an extreme solution to an LP (5)-(9). The SDP and LP can be solved up to additive error of $\epsilon > 0$ in the objective value in $O(n^{6.5} \log(1/\epsilon))$ [Ben-Tal and Nemirovski, 2001] and $O(n^{3.5} \log(1/\epsilon))$ [Schrijver, 1998] time, respectively. The running time of SDP dominates the algorithm both in theory and practice, and is too slow for practical uses for moderate size of $n$.

We propose another algorithm of solving SDP using the multiplicative weight (MW) update method. In theory, our MW takes $O(\frac{1}{\epsilon^2})$ iterations of solving standard PCA, giving a total of $O(\frac{n^3}{\epsilon^2})$ runtime, which may or may not be faster than $O(n^{6.5} \log(1/\epsilon))$ depending on $n, \epsilon$. In practice, however, we observe that after appropriately tuning one parameter in MW, the MW algorithm achieves accuracy $\epsilon < 10^{-5}$ within tens of iterations, and therefore is used to obtain experimental results in this paper. Our MW can handle data of dimension up to a thousand with running time in less than a minute. The details of implementation and analysis of MW method are in Appendix A.

# 6  Experiments

We use two common human-centric data sets for our experiments. The first one is labeled faces in the wild (LFW) [Huang et al., 2007], the second is the Default Credit data set [Yeh and Lien, 2009]. We preprocess all data to have its mean at the origin. For the LFW data, we normalized each pixel value by $\frac{1}{255}$. The gender information for LFW was taken from Afifi and Abdelhamed [2017], who manually verified the correctness of these labels. For the credit data, since different attributes are measurements of incomparable units, we normalized the variance of each attribute to be equal to 1.

**Results**  We focus on projections into relatively few dimensions, as those are used ubiquitously in early phases of data exploration. As we already saw in Figure 1 left, at lower dimensions, there is a noticeable gap between PCA's average reconstruction error for men and women on the LFW data set. This gap is at the scale of up to 10% of the total reconstruction error when we project to 20 dimensions. This still holds when we subsample male and female faces with equal probability from the data set, and so men and women have equal magnitude in the objective function of PCA (Figure 1 right).

Figure 3 shows the average reconstruction error of each population (Male/Female, Higher/Lower education) as the result of running vanilla PCA and Fair PCA on LFW and Credit data. As we expect,

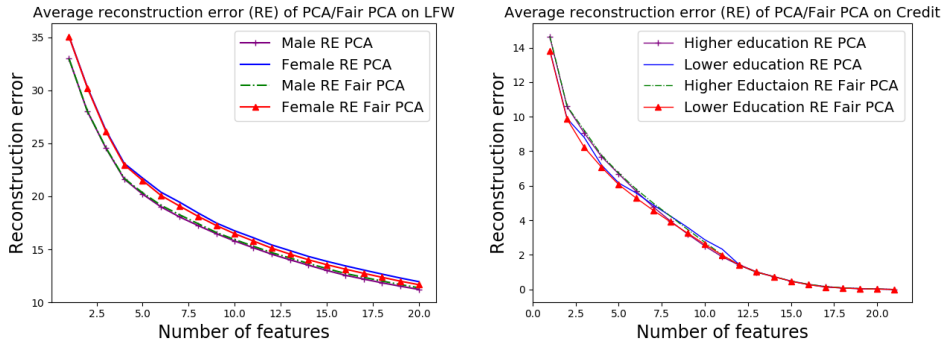

Figure 3: Reconstruction error of PCA/Fair PCA on LFW and the Default Credit data set.

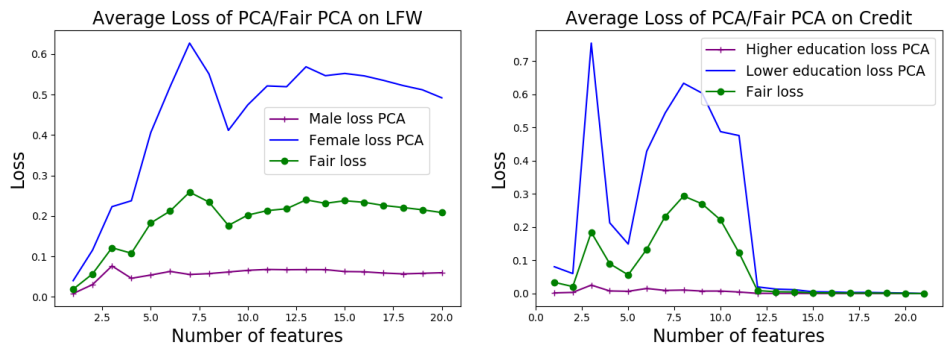

Figure 4: Loss of PCA/Fair PCA on LFW and the Default Credit data set.

as the number of dimensions increase, the average reconstruction error of every population decreases. For LFW, the original data is in 1764 dimensions (42×42 images), therefore, at 20 dimensions we still see a considerable reconstruction error. For the Credit data, we see that at 21 dimensions, the average reconstruction error of both populations reach 0, as this data originally lies in 21 dimensions. In order to see how fair are each of these methods, we need to zoom in further and look at the average loss of populations.

Figure 4 shows the average loss of each population as the result of applying vanilla PCA and Fair PCA on both data sets. Note that at the optimal solution of Fair PCA, the average loss of two populations are the same, therefore we have one line for "Fair loss". We observe that PCA suffers much higher average loss for female faces than male faces. After running fair PCA, we observe that the average loss for fair PCA is relatively in the middle of the average loss for male and female. So, there is improvement in terms of the female average loss which comes with a cost in terms of male average loss. Similar observation holds for the Credit data set. In this context, it appears there is some cost to optimizing for the less well represented population in terms of the better-represented population.

# 7 Future work

This work is far from a complete study of when and how dimensionality reduction might help or hurt the fair treatment of different populations. Several concrete theoretical questions remain using our framework. What is the complexity of optimizing the fairness objective? Is it NP-hard, even for $d = 1$? Our work naturally extends to $k$ predefined subgroups rather than just 2, where the number of additional dimensions our algorithm uses is $k - 1$. Are these additional dimensions necessary for computational efficiency?

In a broader sense, this work aims to point out another way in which standard ML techniques might introduce unfair treatment of some subpopulation. Further work in this vein will likely prove very enlightening.

## Acknowledgements

This work was supported in part by NSF awards CCF-1563838, CCF-1717349, and CCF-1717947.

## Footnotes

[1]Lipton et al. [2017] has asked whether equal treatment requires different models for two groups.

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
