[Supplementary Material · supplementary.pdf]

# A   Improved runtime of semi-definite relaxation by multiplicative weight update method

In this section, we show the multiplicative weight (MW) algorithm and runtime analysis to solve the fair PCA relaxation in two groups for $n \times n$ matrix up to $\epsilon$ additive error in $O(\frac{1}{\epsilon^2})$ iterations of solving a standard PCA, such as Singular Value Decomposition (SVD). Because SVD takes $O(n^3)$ time, the SDP relaxation (4) for two groups can be solved in $O(\frac{n^3}{\epsilon^2})$. Comparing to $O(n^{6.5} \log(1/\epsilon))$ runtime of an SDP solver that is commonly implemented with the interior point method [Ben-Tal and Nemirovski, 2001], our algorithm may be faster or slower depending on $n, \epsilon$. In practice, however, we tune the parameter of MW algorithm much more aggressively than in theory, and often take the last iterate solution of MW rather the average when the last iterate performs better, which gives a much faster convergence rate. Our runs of MW show that MW converges in at most 10-20 iterations. Therefore, we use MW to implement our fair PCA algorithm. We note at the conclusion of this section that the algorithm and analysis can be extended to solving fair PCA in $k$ groups up to additive error $\epsilon$ in $O(\frac{\log k}{\epsilon^2})$ iterations.

Technically, the number of iterations for $k$ groups is $O(\frac{W^2 \log k}{\epsilon^2})$, where $W$ is the width of the problem, as defined in Arora et al. [2012]. $W$ can usually be bounded by the maximum number of input or the optimal objective value. For our purpose, if the total variance of input data over all dimension is $L$, then the width $W$ is at most $L$. For simplicity, we assume $L \leq 1$ (e.g. by normalization in prepossessing step), hence obtaining the $O(\frac{\log k}{\epsilon^2})$ bound on number of iterations.

We first present an algorithmic framework and the corresponding analysis in the next two subsections, and later apply those results to our specific setting of solving the SDP (4) from fair PCA problem. The previous work by Arora et al. [2012] shows how we may solve a feasibility problem of an LP using MW technique. Our main theoretical contribution is to propose and analyze the optimization counterpart of the feasibility problem, and the MW algorithm we need to solve such problem. The MW we develop fits more seamlessly into our fair PCA setting and simplifies the algorithm to be implemented for solving the SDP (4).

## A.1   Problem setup and oracle access

We first formulate the feasibility problem and its optimization counterpart in this section. The previous and new MW algorithms and their analysis are presented in the following Section A.2.

### A.1.1   Previous work: multiplicative weight on feasibility problem

**Problem.**   As in Arora et al. [2012], we are given $A \in \mathbb{R}^{m \times n}$ as an $m \times n$ real matrix, $x \in \mathbb{R}^n, b \in \mathbb{R}^m$, and $\mathcal{P}$ as a convex set in $\mathbb{R}^n$, and the goal is to check the feasibility problem

$$\exists? x \in \mathcal{P} \ : \ Ax \geq b \tag{11}$$

by giving a feasible $x \in \mathcal{P}$ or correctly deciding that such $x$ does not exist.

**Oracle Access.**   We assume the existence if an oracle that, given any probability vector $p \in \Delta_m$ over $m$ constraints of (11), correctly answers a single-constraint problem

$$\exists? x \in \mathcal{P} \ : \ p^\top Ax \geq p^\top b \tag{12}$$

by giving a feasible $x \in \mathcal{P}$ or correctly deciding that such $x$ does not exist. We may think of (12) as a weighted version of (11), with weights on each constraint $i \in [m]$ being $p_i$.

As (12) consists of only one constraint, solving (12) is much easier than (11) in many settings. For example, in our PCA setting, solving (4) directly is non-trivial, but the weighted version (12) is a standard PCA problem: we weight each group $A, B$ based on $p$, and then apply a PCA algorithm (Singular Value Decomposition) on the sum of two weighted groups. The solution gives an optimal value of $p^\top Ax - p^\top b$ in (12). More details of application in fair PCA settings are in Sec. A.3

### A.1.2   New setting: multiplicative weight on optimization problem

**Problem.**   The previous work gives an MW framework for the feasibility question. Here we propose an optimization framework, which asks for the best $x \in \mathcal{P}$ rather than an existence of $x \in \mathcal{P}$.

The optimization framework can be formally stated as, given $A \in \mathbb{R}^{m \times n}$ as an $m \times n$ real matrix, $x \in \mathbb{R}^n, b \in \mathbb{R}^m$, and $\mathcal{P}$ as a convex set in $\mathbb{R}^n$, we need to solve

$$\min z \; : \; Ax - b + z \cdot \mathbf{1} \geq 0, \text{ s.t. } x \in \mathcal{P} \tag{13}$$

where $\mathbf{1}$ denotes the $m \times 1$ vector with entries 1. Denote $z^*$ the optimum of (13).

With the same type of oracle access, we may run (11) for $O(\log \frac{n}{\epsilon})$ iterations to do binary search for the correct value of optimum $z^*$ up to an additive error $\epsilon$. However, our main contribution is to modify the previous multiplicative weight algorithm and the definition of the oracle to solve (13) without guessing the optimum $z^*$. This improves the runtime slightly (reduce the $\log(n/\epsilon)$ factor) and simplifies the algorithm.

**Feasibility Oracle Access.** We assume the existence of an oracle that, given any probability vector $p \in \Delta_m$ over $m$ constraints of (13), correctly answers a single-constraint problem

$$\text{Find } x \in \mathcal{P} \; : \; p^\top Ax - p^\top b + z^* \geq 0 \tag{14}$$

There is always such $x$ because multiplying (13) on the left by $p^\top$ shows that one of such $x$ is the optimum $x^*$ of (13). However, finding one may not be as trivial as asserting problem's feasibility. In general, (14) can be tricky to solve since we do not yet know the value of $z^*$.

**Optimization Oracle Access.** We define the oracle that, given $p \in \Delta_m$ over $m$ constraints of (13), correctly answers one maximizer of

$$\min z \; : \; p^\top Ax - p^\top b + z \geq 0, \text{ s.t. } x \in \mathcal{P} \tag{15}$$

which is stronger than and is sufficient to solve (14). This is because $x^*$ of (13) is one feasible $x$ to (15), so the optimum $\hat{z}$ of (15) is at most $z^*$. Therefore, the optimum $x$ by (15) can be a feasible solution to (14). In many setting, because (14) is only one-constraint problem, it is possible to solve the optimization version (15) instead. For example, in our fair PCA on two groups setting, we can solve the (15) by standard PCA on the union of two groups after an appropriate weighting on each group. More details of application in fair PCA settings are in Section A.3.

## A.2 Algorithm and Analysis

The line of proof follows similarly from Arora et al. [2012]. We first state the technical property that the oracle satisfies in our optimization framework, then show how to use that property to bound the number of iterations. We fix $A \in \mathbb{R}^{m \times n}$ as an $m \times n$ real matrix, $x \in \mathbb{R}^n, b \in \mathbb{R}^m$, and $\mathcal{P}$ is a convex set in $\mathbb{R}^n$

**Definition A.1.** *(analogous to Arora et al. [2012]) An $(\ell, \rho)$-bounded* **oracle** *for parameter $0 \leq \ell \leq \rho$ is an algorithm which, given $p \in \Delta_m$, solve (14). Also, there is a fixed $I \subseteq [m]$ (i.e. fixed across all possible $p \in \Delta_m$) of constraints such that for all $x \in \mathcal{P}$ output by this algorithm,*

$$\forall i \in I : \; A_i x - b_i + z^* \in [-\ell, \rho] \tag{16}$$
$$\forall i \notin I : \; A_i x - b_i + z^* \in [-\rho, \ell] \tag{17}$$

Note that even though we do not know $z^*$, if we know the range of $A_i x - b_i$ for all $i$, we can bound the range of $z^*$. Therefore, we can still find a useful $\ell, \rho$ that an oracle satisfies.

Now we are ready to state the main result of this section: that we may solve the optimization version by multiplicative update as quickly as solving the feasibility version of the problem.

**Theorem A.2.** *Let $\epsilon > 0$ be given. Suppose there exists $(\ell, \rho)$-bounded* **oracle** *and $\ell \geq \epsilon/4$ to solving (14). Then there exists an algorithm that solves (13) up to additive error $\epsilon$, i.e. outputs $x \in \mathcal{P}$ such that*

$$Ax - b + z^* \cdot \mathbf{1} \geq -\epsilon \tag{18}$$

*The algorithm calls* **oracle** *$O(\ell \rho \log(m)/\epsilon^2)$ times and has additional $O(m)$ time per call.*

*Proof.* The proof follows similarly as Theorem 3.3 in Arora et al. [2012], but we include details here for completeness. The algorithm is multiplicative update in nature, as in equation (2.1) of Arora et al. [2012]. The algorithm starts with uniform $p^0 \in \Delta_m$ over $m$ constraints. Each step the algorithm asks

the **oracle** with input $p^t$ and receive $x^t \in \mathcal{P}$. We use the loss vector $m^t = \frac{1}{\rho}(Ax^t - b)$ to update the weight $p^t$ for the next step with learning rate $\eta$. After $T$ iterations (which will be specified later), the algorithm outputs $\bar{x} = \frac{1}{T}\sum_{t=1}^{T} x^t$.

Note that using either the loss $\frac{1}{\rho}(Ax^t - b + z^*)$ and $\frac{1}{\rho}(Ax^t - b)$ behaves the same algorithmically due to the renormalization step on the vector $(p_i^t)_{i=1}^{m}$. Therefore, just for analysis, we use a hypothetical loss $m^t = \frac{1}{\rho}(Ax^t - b + z^*)$ to update $p^t$ (this loss can't be used algorithmically since we do not know $z^*$). By Theorem 2.1 in Arora et al. [2012], for each constraint $i \in [m]$ and all $\eta \leq 1/2$,

$$\sum_{t=1}^{T} m^t \cdot p^t \leq \sum_{t=1}^{T} m_i^t + \eta \sum_{t=1}^{T} |m_i^t| + \frac{\log m}{\eta}$$

$$= \frac{1}{\rho}\sum_{t=1}^{T}(A_i x^t - b_i + z^*) + \frac{\eta}{\rho}\sum_{t=1}^{T}|A_i x^t - b_i + z^*| + \frac{\log m}{\eta} \qquad (19)$$

By property (14) of the **oracle**,

$$\sum_{t=1}^{T} m^t \cdot p^t = \frac{1}{\rho}\sum_{t=1}^{T}\left((p^t)^\top(Ax^t - b) + z^*\right) \geq 0 \qquad (20)$$

We now split into two cases. If $i \in I$, then (19) and (20) imply

$$0 \leq \frac{1+\eta}{\rho}\sum_{t=1}^{T}(A_i x^t - b_i + z^*) + \frac{2\eta}{\rho}\sum_{t:A_i x^t - b_i < 0}|A_i x^t - b_i + z^*| + \frac{\log m}{\eta}$$

$$\leq \frac{1+\eta}{\rho}T(A_i\bar{x} - b_i + z^*) + \frac{2\eta}{\rho}T\ell + \frac{\log n}{\eta}$$

Multiplying the last inequality by $\frac{\rho}{T}$ and rearranging terms, we have

$$0 \leq (1+\eta)(A_i\bar{x} - b_i + z^*) + 2\eta\ell + \frac{\rho\log m}{T\eta} \qquad (21)$$

If $i \notin I$, then (19) and (20) imply

$$0 \leq \frac{1-\eta}{\rho}\sum_{t=1}^{T}(A_i x^t - b_i + z^*) + \frac{2\eta}{\rho}\sum_{t:A_i x^t - b_i > 0}|A_i x^t - b_i + z^*| + \frac{\log m}{\eta}$$

$$\leq \frac{1-\eta}{\rho}T(A_i\bar{x} - b_i) + \frac{2\eta}{\rho}T\ell + \frac{\log n}{\eta}$$

Multiplying inequality by $\frac{\rho}{T}$ and rearranging terms, we have

$$0 \leq (1-\eta)(A_i\bar{x} - b_i + z^*) + 2\eta\ell + \frac{\rho\log m}{T\eta} \qquad (22)$$

To use (21) and (22) to show that $A_i\bar{x} - b_i + z^*$ is close to 0 simultaneously for two cases, pick $\eta = \frac{\epsilon}{8\ell}$ (note that $\eta \leq 1/2$ by requiring $\ell \geq \epsilon/4$, so we may apply Theorem 2.1 in Arora et al. [2012]). Then for all $T \geq \frac{4\rho\log(m)}{\epsilon\eta} = \frac{32\ell\rho\log(m)}{\epsilon^2}$, we have

$$2\eta\ell + \frac{\rho\log m}{T\eta} \leq \frac{\epsilon}{4} + \frac{\epsilon}{4} = \frac{\epsilon}{2} \qquad (23)$$

Hence, (21) implies

$$0 \leq (1+\eta)(A_i\bar{x} - b_i + z^*) + \frac{\epsilon}{2} \Rightarrow A_i\bar{x} - b_i + z^* \geq -\frac{\epsilon}{2} \qquad (24)$$

and (22) implies

$$0 \leq (1-\eta)(A_i\bar{x} - b_i + z^*) + \frac{\epsilon}{2} \Rightarrow A_i\bar{x} - b_i + z^* \geq -\epsilon \qquad (25)$$

using the fact that $\eta \leq 1/2$. $\qquad\qquad\square$

## A.3 Application of multiplicative update method to the fair PCA problem

In this section, we apply MW results for solving LP to solve the SDP relaxation (4) of fair PCA.

**LP formulation of fair PCA relaxation.** The SDP relaxation (4) of fair PCA can be written in the form (13) as an LP with two constraints

$$\min_{P \in \mathcal{P}, z \in \mathbb{R}} z \text{ s.t.} \tag{26}$$

$$z \geq \alpha - \frac{1}{m_1} \langle A^\top A, P \rangle \tag{27}$$

$$z \geq \beta - \frac{1}{m_2} \langle B^\top B, P \rangle \tag{28}$$

for some constants $\alpha, \beta$, where the feasible region of variables is over a set of PSD matrices:

$$\mathcal{P} = \{ M \in \mathbb{R}^{n \times n} : 0 \preceq M \preceq I, \text{tr}(M) \leq d \} \tag{29}$$

We will apply the multiplicative weight algorithm to solve (26)-(28).

**Oracle Access.** First, we present an the oracle in Algorithm 2, which is in the form (15) and therefore can be used to solve (14). As defined in (15), the optimization oracle, given a weight vector $p = (p_1, p_2) \in \Delta_2$, should be able to solve the LP with one weighted constraint obtained from weighting two constraints (27) and (28) by $p$. However, because both constraints involve only dot products of same variable $P$ with constant matrices $A^\top A$ and $B^\top B$, which are linear functions, the weighted constraint will involve the dot product of the same variable $P$ with weighted sum of those constant matrices $\frac{p_1}{m_1} A^\top A + \frac{p_2}{m_2} B^\top B$.

---

**Algorithm 2:** Fair PCA oracle (oracle to Algorithm 3)

**Input** : $p = (p_1, p_2) \in \Delta_2, \alpha, \beta \in \mathbb{R}, A \in \mathbb{R}^{m_1 \times n}, B \in \mathbb{R}^{m_2 \times n}$
**Output** : $\arg\min_{P, z_1, z_2} p_1 z_1 + p_2 z_2$, subject to
$\quad z_1 = \alpha - \frac{1}{m_1} \langle A^\top A, P \rangle,$
$\quad z_2 = \beta - \frac{1}{m_2} \langle B^\top B, P \rangle,$
$\quad P \in \mathcal{P} = \{ M \in \mathbb{R}^{n \times n} : 0 \preceq M \preceq I, \text{Tr}(M) \leq d \}$

1 Set $V \in \mathbb{R}^{n \times d}$ to be the matrix with top $d$ principles components of $\frac{p_1}{m_1} A^\top A + \frac{p_2}{m_2} B^\top B$ as columns;

2 **return** $P^* = VV^\top$, $z_1^* = \alpha - \frac{1}{m_1} \langle A^\top A, P^* \rangle$, $z_2^* = \beta - \frac{1}{m_2} \langle B^\top B, P^* \rangle$;

---

**MW Algorithm.** Our multiplicative weight update algorithm for solving fair PCA relaxation (26)-(28) is presented in Algorithm 3. The algorithm follows exactly from the construction in Theorem A.2. The runtime analysis of our MW Algorithm 3 follows directly from the same theorem.

**Corollary A.3.** *Let $\epsilon > 0$. Algorithm 3 finds a near-optimal (up to additive error of $\epsilon$) solution $P$ to (26)-(28) in $O\left(\frac{1}{\epsilon^2}\right)$ iterations of solving standard PCA, and therefore in $O(\frac{n^3}{\epsilon^2})$ running time.*

*Proof.* We first check that the oracle presented in Algorithm 2 satisfies $(\ell, \rho)$-boundedness and find those parameters. We may normalize the data so that the variances of $\frac{A^\top A}{m_1}$ and $\frac{B^\top B}{m_2}$ are bounded by 1. Therefore, for any PSD matrix $P \preceq I$, we have $\frac{1}{m_1} \langle A^\top A, P \rangle \leq 1$. In addition, in the application to fair PCA setting, we have $\alpha = \frac{\|\widehat{A}\|_F^2}{m_1}$. Hence, $\frac{1}{m_1} \langle A^\top A, P \rangle \leq \alpha$ for any feasible $P \in \mathcal{P} = \{ M \in \mathbb{R}^{n \times n} : 0 \preceq M \preceq I, \text{Tr}(M) \leq d \}$ by the definition of $\widehat{A}$ (recall Definition 3.1). Therefore,

$$0 \leq \alpha - \frac{1}{m_1} \langle A^\top A, P \rangle \leq 1, \forall P \in \mathcal{P} \tag{30}$$

and similarly $\beta - \frac{1}{m_2} \langle B^\top B, P \rangle \in [0, 1]$. Hence, the optimal solution of Algorithm 3 satisfies $z^* \in [0, 1]$. Therefore, the oracle is $(1, 1)$-bounded.

**Algorithm 3:** Multiplicative weight update for fair PCA

---

**Input** : $\alpha, \beta \in \mathbb{R}$, $A \in \mathbb{R}^{m_1 \times n}$, $B \in \mathbb{R}^{m_2 \times n}$, $\eta > 0$, positive integer $T$

**Output** : $\arg\min\limits_{P,z} \; z$, subject to
$$z \geq \alpha - \frac{1}{m_1}\langle A^\top A, P \rangle,$$
$$z \geq \beta - \frac{1}{m_2}\langle B^\top B, P \rangle,$$
$$P \in \mathcal{P} = \{M \in \mathbb{R}^{n \times n} : 0 \preceq M \preceq I, \mathrm{Tr}(M) \leq d\}$$

**1** Initialize $p^0 = (1/2, 1/2)$;

**2 for** $t = 1, \ldots, T$ **do**

**3** $\quad (P_t, m_1^t, m_2^t) \leftarrow \mathbf{oracle}(p^{t-1}, \alpha, \beta, A, B)$;

**4** $\quad \hat{p}_i^t \leftarrow p_i^{t-1} e^{\eta m_i^t}$, for $i = 1, 2$;

**5** $\quad p_i^t \leftarrow \hat{p}_i^t / (\hat{p}_1^t + \hat{p}_2^t)$, for $i = 1, 2$;

**6 end**

**7 return** $P^* = \frac{1}{T}\sum_{t=1}^{T} P_t$ , $z^* = \max\{\alpha - \frac{1}{m_1}\langle A^\top A, P^* \rangle, \beta - \frac{1}{m_2}\langle B^\top B, P^* \rangle\}$

---

Next we analyze the runtime of Algorithm 3. By Theorem A.2, Algorithm 3 calls the oracle $O(1/\epsilon^2)$ times. The bottleneck in an oracle call is solving PCA on the weighted sum of two groups, which takes $O(n^3)$ time. The additional processing time to update the weight is negligible compared to this $O(n^3)$ time for solving PCA. $\qquad\square$

**MW for More Than Two Groups.** Algorithms 2 and 3 can be naturally extended to $k$ groups. Theorem A.2 states that we need $O(\frac{\log k}{\epsilon^2})$ calls to the oracle with additional $O(k)$ time per call (to update the weight for each loss). In each call, we must compute the weighted sum of $k$ matrices of dimension $n \times n$, which takes $O(kn^2)$ arithmetic operations and perform SVD. In natural settings, $k$ is much smaller than $n$, and hence the runtime $O(n^3)$ of SVD in each oracle call will dominate.

# B  Proofs

**Proof of Lemma 4.6**: Since $\mathsf{rank}(U) \leq d$, $\mathsf{rank}(V) \leq d$ and thus $\mathsf{rank}(\begin{bmatrix} A \\ B \end{bmatrix} VV^T) \leq d$. We will first show that $loss(A, AVV^T) \leq loss(A, U_A)$.

**Step 1.** Since $\{v_1, \ldots, v_d\}$ is an orthonormal basis of row space of $U$, for every row of $U_A$, we have that $(U_A)_i = c_i V^T$ for some $c_i \in \mathbb{R}^{1 \times d}$.

**Step 2.** We show if we $c_i \to A_i V$ and consequently substitute the row $(U_A)_i \to A_i VV^T$, the value of $\|A_i - (U_A)_i\|$ decreases.

$$\|A_i - (U_A)_i\|^2 = \|A_i - c_i V^T\|^2 = A_i A_i^T - 2A_i V c_i^T + c_i c_i^T$$

We used the fact that $V^T V = I$. Minimizing the RHS with respect to $c_i$ results in $c_i = A_i V$.

**Step 3.** Step 2 proved that for every $i$, $\|A_i - A_i VV^T\|^2 \leq \|A_i - (U_A)_i\|^2$. Remember that

$$loss(A, U_A) = \|A - U_A\|_F^2 - \|A - \widehat{A}\|_F^2 = \sum \|A_i - (U_A)_i\|^2 - \|A - \widehat{A}\|_F^2$$
$$loss(A, AVV^T) = \|A - AVV^T\|_F^2 - \|A - \widehat{A}\|_F^2 = \sum \|A_i - A_i VV^T\|^2 - \|A - \widehat{A}\|_F^2$$

This finished the proof that $loss(A, AVV^T) \leq loss(A, U_A)$. Similarly, we can see that $loss(B, BVV^T) \leq loss(B, U_B)$. Therefore

$$f\left(\begin{bmatrix} A \\ B \end{bmatrix} VV^T\right) = \max\left(\frac{1}{|A|}loss(A, AVV^T), \frac{1}{|B|}loss(B, BVV^T)\right)$$
$$\leq \max\left(\frac{1}{|A|}loss(A, U_A), \frac{1}{|B|}loss(B, U_B)\right) = f(U)$$

□

**Proof of Lemma 4.7**: From Lemma 3.1, we know that there exist a matrix $W_A \in \mathbb{R}^{n \times d}$ such that $W_A^T W_A = I$ and $\widehat{A} = A W_A W_A^T$. Considering this and the fact that $V^T V = I$

$$loss(A, AVV^T) = \|A - AVV^T\|_F^2 - \|A - AW_A W_A^T\|_F^2$$

$$= \sum_i \|A_i - A_i VV^T\|^2 - \|A_i - A_i W_A W_A^T\|^2$$

$$= \sum_i A_i A_i^T - A_i VV^T A_i^T - \left(\sum_i A_i A_i^T - \sum_i A_i W_A W_A^T\right)$$

$$= \sum_i A_i W_A W_A^T A_i^T - \sum_i A_i VV^T A_i^T$$

$$\sum_i A_i W_A W_A^T A_i^T = \sum_i \|A_i W_A\|^2 = \|AW_A\|_F^2 = \|AW_A W_A^T\|_F^2 = \|\widehat{A}\|_F^2$$

$$\sum_i A_i VV^T A_i^T = \sum_i \|A_i V\|^2 = \|AV\|_F^2 = \sum_i \|Av_i\|^2$$

$$\sum_i A_i VV^T A_i^T = \sum_i \|A_i V\|^2 = \|AV\|_F^2 = \text{Tr}(V^T A^T A V) = \text{Tr}(VV^T A^T A) = \langle A^T A, VV^T \rangle$$

Therefore $loss(A, AVV^T) = \|\widehat{A}\|_F^2 - \sum_{i=1}^d \|Av_i\|^2 = \|\widehat{A}\|_F^2 - \langle A^T A, VV^T \rangle$.

$$\|A - AVV^T\|_F^2 = \sum_i \|A_i - A_i VV^T\|^2 = \sum_i A_i A_i^T - \sum_i A_i VV^T A_i^T$$

$$= \|A\|_F^2 - \sum_i \|Av_i\|^2 = \|A\|_F^2 - \|AV\|_F^2$$

□

**Proof of Lemma 4.8**:

We prove that the value of function $g_A$ at its local minima is equal to its value at its global minimum, which we know is the subspace spanned by a top $d$ eigenvectors of $A^T A$. More precisely, we prove: Let $\{v_1, \ldots, v_n\}$ be an orthonormal basis of eigenvectors of $A^T A$ with corresponding eigenvalues $\lambda_1 \geq \ldots \geq \lambda_n$ where ties are broken arbitrarily. Let $V^*$ be the subspace spanned by $\{v_1, \ldots, v_d\}$ and let $U$ be some $d$-dimensional subspace s.t. $g_A(U) > g_A(V^*)$. There is a continuous path from $U$ to $V^*$ s.t. the value of $g_A$ monotonically decreases for every $d$-dimensional subspace on the path.

Before starting the proof, we will make a couple of notes which would be used throughout the proof. First note that $g_A(V)$ is well-defined i.e., the value of $g_A(V)$ is only a function of the subspace $V$. More precisely, $g_A(V)$ is invariant with respect to different choices of orthonormal basis of the subspace $V$. Second, given Lemma 4.7, $g_A(V) = \|A\|_F^2 - \sum_i \|Av_i\|^2$. Therefore, proving that $g_A(V)$ is decreasing is equivalent to proving that $\sum_i \|Av_i\|^2$ is increasing as a function of any choice of orthonormal basis of the subspaces on the path.

$g_A(U) > g_A(V^*)$ therefore $U \neq V^*$. Let $k$ be the smallest index such that $v_k \notin U$. Extend $\{v_1, \ldots, v_{k-1}\}$ to an orthonormal basis of $U$: $\{v_1, \ldots, v_{k-1}, v_k', \ldots, v_d'\}$. Let $q \geq k$ be the smallest index such that $\|Av_q\|^2 > \|Av_q'\|^2$. Such an index $q$ must exist given that $g_A(U) > g_A(V^*)$. Without loss of generality we can assume that $q = 1$. Therefore, we assume that $v_1$, the top eigenvector of $A^T A$, is not in $U$ and that it strictly maximizes the function $\|Au\|^2$ over the space of unit vectors $u$. Specifically, for any unit vector $u \in U$, $\|Au\|^2 < \|Av_1\|^2 = \lambda_1$. Let $v_1 = \sqrt{1 - a^2} z_1 + a z_2$ where $z_1 \in U$ and $z_2 \perp U$, $\|z_1\| = \|z_2\| = 1$ i.e., the projection of $v_1$ to $U$ is $\sqrt{1 - a^2} z_1$. We distinguish two cases:

**Case $z_1 = 0$.** $v_1 \perp U$. Let $w = \sqrt{1 - \epsilon^2} u_1 + \epsilon v_1$. $\|w\| = 1$. Note that $\{w, u_2, \ldots, u_d\}$ is an orthonormal set of vectors. We set $U_\epsilon = span\{w, u_2, \ldots, u_d\}$. We show that $g_A(U_\epsilon) < g_A(U)$.

Using the formulation of $g$ from Lemma 4.7, we need to show that $\|Aw\|^2 + \|Au_2\|^2 + \ldots + \|Au_d\|^2 > \|Au_1\|^2 + \|Au_2\|^2 + \ldots + \|Au_d\|^2$ or equivalently that $\|Aw\|^2 > \|Au_1\|^2$.

$$
\begin{aligned}
\|Aw\|^2 - \|Au_1\|^2 &= \|A(\sqrt{1-\epsilon^2}u_1 + \epsilon v_1)\|^2 - \|Au_1\|^2 \\
&= (\sqrt{1-\epsilon^2}u_1^T + \epsilon v_1^T)A^T A(\sqrt{1-\epsilon^2}u_1 + \epsilon v_1) - \|Au_1\|^2 \\
&= (1-\epsilon^2)u_1^T A^T Au_1 + \epsilon^2 v_1^T A^T Av_1 + 2\sqrt{1-\epsilon^2}\epsilon u_1^T A^T Av_1 - \|Au_1\|^2 \\
&= (1-\epsilon^2)\|Au_1\|^2 + \epsilon^2 \lambda_1 + 2\epsilon\sqrt{1-\epsilon^2}u_1^T A^T Av_1 - \|Au_1\|^2 \\
&= \epsilon^2(\lambda_1 - \|Au_1\|^2) + 2\epsilon\sqrt{1-\epsilon^2}u_1^T A^T Av_1
\end{aligned}
$$

where $u_1^T A^T Av_1 = u_1^T(\lambda_1 v_1) = \lambda_1 u_1^T v_1 = 0$ since $v_1$ is an eigenvector of $A^T A$ and $v_1 \perp u_1$. This, and considering the fact that $\|Au_1\|^2 < \lambda_1$

$$\|Aw\|^2 - \|Au_1\|^2 = \epsilon^2(\lambda_1 - \|Au_1\|^2) > 0$$

Therefore, $\|Aw\|^2 > \|Au_1\|^2$ and thus $g_A(U_\epsilon) < g_A(U)$.

**Case $z_1 \neq 0$.** Note that $z_2 \neq 0$ either since we picked $v_1 \notin U$. Let's extend $\{z_1\}$ to an orthonormal basis of $U$: $\{z_1, u_2, \ldots, u_k\}$. We will transform $U$ s.t. the resulting subspace $U_1$ is the span of $v_1, u_2, \ldots, u_k$. This can then be repeated orthogonal to $v_1$ till the subspace becomes $V^*$.

For small enough $\epsilon > 0$, consider the unit vector $w = \sqrt{1-\epsilon^2}z_1 + \epsilon z_2$. We will move $U$ to $U_\epsilon := span\{w, u_2, \ldots, u_d\}$. The latter is an orthonormal representation since both $z_1$ and $z_2$ are orthogonal to all of $u_2, \ldots, u_d$ and $w$ is in the span of $z_1, z_2$. We will prove that $g_A(U_\epsilon) < g_A(U)$. Given Lemma 4.7, since the chosen orthonormal basis of these two subspaces differ only in $w$ and $z_1$, it suffices to show that $\|Aw\|^2 > \|Az_1\|^2$. We can write

$$
\begin{aligned}
w &= \left(\sqrt{1-\epsilon^2} - \frac{\epsilon\sqrt{1-a^2}}{a}\right)z_1 + \frac{\epsilon}{a}\left(\sqrt{1-a^2}z_1 + az_2\right) \\
&= \left(\sqrt{1-\epsilon^2} - \frac{\epsilon\sqrt{1-a^2}}{a}\right)z_1 + \frac{\epsilon}{a}v_1.
\end{aligned}
$$

Thus, noting that $A^T Av_1 = \lambda_1 v_1$ ($v_1$ is an eigenvector with eigenvalue $\lambda_1$) and $z_1^T v_1 = \sqrt{1-a^2}$,

$\|Aw\|^2$

$$
= \left(\sqrt{1-\epsilon^2} - \frac{\epsilon\sqrt{1-a^2}}{a}\right)^2 \|Az_1\|^2 + \frac{\epsilon^2}{a^2}\|Av_1\|^2 + 2\frac{\epsilon}{a}\left(\sqrt{1-\epsilon^2} - \frac{\epsilon\sqrt{1-a^2}}{a}\right)z_1^T A^T Av_1
$$

$$
= \left(1 - \epsilon^2 + \frac{\epsilon^2(1-a^2)}{a^2} - 2\frac{\epsilon\sqrt{(1-\epsilon^2)(1-a^2)}}{a}\right)\|Az_1\|^2 + \frac{\epsilon^2}{a^2}\lambda_1
$$

$$
+ 2\frac{\epsilon}{a}\left(\sqrt{1-\epsilon^2} - \frac{\epsilon\sqrt{1-a^2}}{a}\right)\lambda_1 z_1^T v_1
$$

$$
= \left(1 - 2\epsilon^2 + \frac{\epsilon^2}{a^2} - 2\frac{\epsilon\sqrt{(1-\epsilon^2)(1-a^2)}}{a}\right)\|Az_1\|^2
$$

$$
+ \left(\frac{\epsilon^2}{a^2} + 2\frac{\epsilon\sqrt{(1-\epsilon^2)(1-a^2)}}{a} - 2\frac{\epsilon^2(1-a^2)}{a^2}\right)\lambda_1
$$

$$
= \|Az_1\|^2 + (\lambda_1 - \|Az_1\|^2)\left(2\frac{\epsilon\sqrt{(1-\epsilon^2)(1-a^2)}}{a} + 2\epsilon^2 - \frac{\epsilon^2}{a^2}\right) > \|Az_1\|^2
$$

The last inequality follows since $\lambda_1 > \|Az_1\|^2$ and we can choose $0 < \epsilon < \frac{1}{1+C}$ for $C = 4a^2(1-a^2)$ so that $2\frac{\epsilon\sqrt{(1-\epsilon^2)(1-a^2)}}{a} > \frac{\epsilon^2}{a^2}$ . Thus, $\|Aw\|^2 > \|Az_1\|^2$ and therefore $g_A(U_\epsilon) < g_A(U)$. $\qquad\square$