[Reviews · NeurIPS 2018]

Reviewer 1



The authors propose a method for considering fairness across groups defined by protected attributes when doing PCA. I like how the authors don't stop at the obvious objective with an equal number of dimensions per group, but think deeper about the problem and come to the formulation they come to. I also think the contextualization to the recent literature, including allocative/representational is good. The math, which results in a semidefinite program appears to be correct. In the related work section, it would be nice to expand beyond Zemel et al. (2013) to newer pieces of work in the same vein: H. Edwards and A. Storkey, “Censoring representations with an adversary,” in Proceedings of the International Conference on Learning Representations, San Juan, Puerto Rico, May 2016. A. Beutel, J. Chen, Z. Zhao, and E. H. Chi, “Data decisions and theoretical implications when adversarially learning fair representations,” in Proceedings of the Workshop on Fairness, Accountability, and Transparency in Machine Learning, Halifax, Canada, Aug. 2017. F. P. Calmon, D. Wei, B. Vinzamuri, K. Natesan Ramamurthy, and K. R. Varshney, “Optimized pre-processing for discrimination prevention,” in Advances in Neural Information Processing Systems 30, Long Beach, USA, Dec. 2017, pp. 3992–4001. B. H. Zhang, B. Lemoine, and M. Mitchell, “Mitigating unwanted biases with adversarial learning,” in Proceedings of the AAAI/ACM Conference on Artificial Intelligence, Ethics, and Society, New Orleans, USA, Feb. 2018. D. Madras, E. Creager, T. Pitassi, and R. Zemel, “Learning adversarially fair and transferable representations,” arXiv:1802.06309, Feb. 2018. Remark 4.1 can be accompanied by reference to Z. C. Lipton, A. Chouldechova, and J. McAuley, “Does mitigating ML’s impact disparity require treatment disparity?” arXiv:1711.07076, Feb. 2018. --- I would have liked the "practical side" future work done in this paper itself. --- Has many typing, typesetting, etc. problems that should be fixed.

Reviewer 2



The authors motivate the necessity of novel dimensionality reduction techniques that address the notion of fairness. The fairness is understood as balanced reconstruction errors for each of the two groups rather than an average global reconstruction error. Thus the authors propose a novel formulation of the dimensionality reduction problem and propose a solution to it with analytical performance guarantees. I appreciate that the authors motivate why they search for one fair projection instead of trivial finding of different projects per groups. I have two remarks to this work: The first one is about the relevance of class-conditional projections like parametric and non-parementic versions of LDA by Fugunaga. They explicitly account for within class and between class covanriances (but with a different optimization criteria of course). Intuitively, a similar approach would allow for a closed-form solution. Perhaps non-negative SVD shall be covered in the related work too as a different, but related problem, i.e. finding projections that optimize for better representations of individual groups of data points (whem classes/sensitive groups are not predefined). The second part is that the title is not obvious to interpret. I am not sure what is considered as a price of the fair PCA. I though that this would be a loss in average reconstruction error due to balancing of per group reconstruction errors. Minor: there are small presentation issues, e.g. Figure ?? === I read the authors feedback

Reviewer 3



The manuscript proposes a dimensionality reduction method called “fair PCA”. The proposed study is based on the observation that, in a data model containing more than one data category (“population” as called by authors), the projection learnt by PCA may yield different reconstruction errors for different populations. This may impair the performance of machine learning algorithms that have access to dimensionality-reduced data obtained via PCA. To address this problem, the authors propose a variant of the PCA algorithm that minimizes the total deviation between the error of the learnt projection and the error of the optimal projection for each population. Quality: The paper is based on an interesting idea with an interesting motivation. The technical content of the paper is of satisfactory depth. The proposed work can find usage in many applications. A few comments and questions: - A setting with only two populations is considered in the theoretical results and the proposed algorithm. Is it possible to extend these results to a setting with multiple populations? In particular, is it easy to suggest a multiple-population version of Theorem 4.5? - The experimental evaluation of the proposed algorithm is somewhat limited. It would be very interesting to see whether the proposed dimensionality reduction method improves the performance of machine learning algorithms when used for preprocessing the data containing two/multiple populations. - Please have a complete pass over the paper to check the language. There are several sentences with typos and mistakes. Clarity: The presentation of the study is of satisfactory clarity in general. However, please address the following issues: - The discussion in page 2, lines 62-69 is hard to follow. It would be good to paraphrase with more clear wording. - Please define the notation [A/B] in line 5 of Algorithm 1. - In the first plot in Figure 2, the loss seems to increase with the number of dimensions. This contradicts the expectation that the loss should decay to 0 in the long term as the number of dimensions increases, since the approximation error of both PCA and fair PCA would approach 0. How do the authors explain this behavior? Originality: As far as I know, the work is based on an original idea and problem formulation. Significance: The proposed study has solid content and may find useful application in many machine learning problems.